# Mapping Urban Green Spaces at the Metropolitan Level Using Very High Resolution Satellite Imagery and Deep Learning Techniques for Semantic Segmentation

Roberto E. Huerta [1], Fabiola D. Yépez [1,*], Diego F. Lozano-García [2], Víctor H. Guerra Cobián [1], Adrián L. Ferriño Fierro [1], Héctor de León Gómez [1], Ricardo A. Cavazos González [1] and Adriana Vargas-Martínez [2]

1. Facultad de Ingeniería Civil, Universidad Autónoma de Nuevo León, San Nicolás de los Garza 66455, Mexico; roberto.huertagc@uanl.edu.mx (R.E.H.); victor.guerracb@uanl.edu.mx (V.H.G.C.); adrian.ferrinofr@uanl.edu.mx (A.L.F.F.); hector.deleongm@uanl.edu.mx (H.d.L.G.); ricardo.cavazosgzz@uanl.edu.mx (R.A.C.G.)
2. Escuela de Ingeniería y Ciencias, Tecnológico de Monterrey, Ave. Eugenio Garza Sada 2501, Monterrey 64849, Mexico; dflozano@tec.mx (D.F.L.-G.); adriana.vargas.mtz@tec.mx (A.V.M.)
* Correspondence: fabiola.yepezrn@uanl.edu.mx; Tel.: +52-8114-424-4000 (ext. 7258)

**Abstract:** Urban green spaces (UGSs) provide essential environmental services for the well-being of ecosystems and society. Due to the constant environmental, social, and economic transformations of cities, UGSs pose new challenges for management, particularly in fast-growing metropolitan areas. With technological advancement and the evolution of deep learning, it is possible to optimize the acquisition of UGS inventories through the detection of geometric patterns present in satellite imagery. This research evaluates two deep learning model techniques for semantic segmentation of UGS polygons with the use of different convolutional neural network encoders on the U-Net architecture and very high resolution (VHR) imagery to obtain updated information on UGS polygons at the metropolitan area level. The best model yielded a Dice coefficient of 0.57, IoU of 0.75, recall of 0.80, and kappa coefficient of 0.94 with an overall accuracy of 0.97, which reflects a reliable performance of the network in detecting patterns that make up the varied geometry of UGSs. A complete database of UGS polygons was quantified and categorized by types with location and delimited by municipality, allowing for the standardization of the information at the metropolitan level, which will be useful for comparative analysis with a homogenized and updated database. This is of particular interest to urban planners and UGS decision-makers.

**Keywords:** neural networks; urban vegetation; urban open spaces; Monterrey Metropolitan Area; sustainable development

## 1. Introduction

Urban green spaces (UGSs) face significant challenges due to rapid urbanization and climate change [1]. UGSs are crucial in order to safeguard the quality of urban life [2]. City managers are urged to integrate UGSs in urban development plans [3]. The conservation of ecosystem services of UGSs can mitigate the impacts of urban development; can reduce ecological debts; and is the simplest, fastest, and most effective way to ameliorate important challenges in cities, such as urban heat islands and air pollution [4–6]. UGSs enhance resilience, health, and quality of life of citizens, especially benefiting those with high accessibility. UGS accessibility is a crucial aspect of sustainable urban planning [7] and social justice [8].

UGS survey data are not commonly updated or freely accessible to local users. There is a need for uniform and spatially explicit inventories of existing UGSs [9] and the quantification of their proximity services [10]. One of the most critical functions of the UGSs within cities is the provision of essential environmental services, as it is related to human well-being [11,12]. UGSs contribute to the reduction of harmful effects that cause cardiovascular,

respiratory, and metabolic diseases [13], and they mitigate the stress caused by increases in temperature and noise levels [14]. Additionally, UGSs promote physical activity and social interaction, improving the physical and mental health of residents who use these facilities [15].

The constant environmental, social, and economic transformations of cities make the management of UGSs a major challenge for government administrations in metropolitan areas with extensive and rapid urban development [16,17]. Depending on the management of the UGSs, negative or positive effects can be promoted, from the destruction of these spaces [18] to promoting adequate conditions for management and maintenance [19]. Inventories of UGSs allow the monitoring of their status and help provide guidelines for the development of adequate management strategies [20].

UGSs exist in diverse shapes, sizes, vegetation covers, and types (i.e., park, residential garden, median strip, square, and roundabout) [21,22]. Traditional methods to obtain polygons of UGSs have relied on the visual interpretation of aerial imagery, remotely sensed data interpretation, and manual digitalization [23]. Similar to tree inventories and other natural elements in UGSs, data collection methods are intensive and involve manual measurements of dendometric parameters in the field. These methods are time-consuming and costly [24]. The integration of remote sensing and geographic information systems for mapping and monitoring UGSs has been advantageous, as this reduces the resources required by traditional methods [25–27]. Moreover, the use of these techniques and frameworks allows the computation of vegetation indices that highlight vegetation properties such as vegetation cover and vigor [28].

Methods based on different machine learning algorithms, including decision tree [29], maximum likelihood classification [30], random forest, and support vector regression [31], have been used to map UGSs. Other authors propose the use of object-based image analysis, which takes advantage of both the spectral and contextual information of the classifying objects [32]. With technological advancement and the evolution of deep learning, optimization of the acquisition of UGS inventories is possible through the detection of spectral and geometric patterns available in satellite imagery [33]. Convolutional neural networks (CNNs) have performed well at high-level vision tasks, such as image classification, object detection, and semantic segmentation [34]. A combination of multitemporal MODIS and Landsat-7 imagery was used to classify UGSs in Mumbai metropolitan area in India [35]. The results indicate that for over 15 years, the overall UGSs were reduced to 50%. Other authors analyzed four different methods of classifying UGSs: support vector machine, random forest, artificial neural networks, and naïve Bayes classifier [36]. They found that support vector machines produce higher accuracy classifications in a short amount of time. Multitemporal high-resolution imagery was employed to map open spaces in Kampala, Uganda, with the use of a cloud computation method and machine learning that combined nine base classifiers [37]. The results produced a map of open spaces with an 88% classification accuracy. A deep learning classification based upon a high-resolution network (HRNet) method of high-resolution GaoFen-2 imagery was used for the city of Beijing, China, indicating that the HRNet combined with phenological analysis significantly improved the classification of UGSs [38].

CNNs are convenient models for semantic segmentation because they produce hierarchies that help determine low-, medium-, and high-level characteristics [39,40]. These models are automatically trained using previously labeled input information, and they produce class identification results [41]. With the use of labeled samples, a network can update its weights until it obtains a proper mapping of the inputs and a minimal loss [42].

Due to the absence of a dense layer, the use of fully convolutional networks (FCNs) allows the generation of outputs in which each pixel has a classification according to the input information [43]. Based on the FCN model, the U-Net architecture uses the same principle and considers a symmetric encoder–decoder composition. This process first reduces the size and increases the number of bands of the training images and their activation maps generated in each layer of the network to subsequently carry out the opposite

process considering information from the encoder in the segmentation of fine details [44]. These types of networks have achieved wide success with state-of-the-art results for a wide variety of problems from medical applications [45,46] to their employment in remote sensing for road [47] and building extractions [48], as well as land cover classification [49], but they have not been used to make many advances in the UGS area.

Detailed geometric information on UGSs is typically presented as a shapefile that is not updated frequently; therefore, it does not reflect changes occurring due to rapid urban development processes. Additionally, spatial data or information about the availability of UGSs is not generally accessible to urban residents [50], restricting their use. The need to improve and make available geospatial data of green and public spaces is recognized by the United Nations sustainable development agenda as it helps to create more inclusive, safe, resilient, sustainable cities [51]. The generation of UGS inventories in conjunction with other public space inventories aid in the calculation of Sustainable Development Goal (SDG) 11.7.1, i.e., quantifying the average share of green and public spaces in cities. This allows for the obtainment of SGD 11.7, which is to "provide universal access to safe, inclusive and accessible, green and public spaces". It is therefore necessary to later relate the information available in digitization, vectorization, and computation to demographic data to generate accessibility maps [52,53]. This study evaluates two deep learning model techniques for semantic segmentation of UGS polygons. The process involves different convolutional neural network encoders on the U-Net architecture with the use of three-band compositions of very high resolution (VHR) satellite imagery channels and vegetation indices as input data. This precise and updated data collection and new UGS cartography at the metropolitan level would improve the understanding of connectivity and accessibility of UGSs as a basis for management and decision-making for land use in urban areas.

## 2. Materials and Methods

### 2.1. Study Area

The study area chosen to test this method was the Monterrey Metropolitan Area (MMA) (Figure 1), located at the coordinates 25°40′00″ N 100°18′00″ W. It has a total area of 6687.10 km$^2$ of which 27.57% is built-up area. The MMA is comprised of Monterrey, the capital of the State of Nuevo Leon, and 11 surrounding municipalities [54]. Its population, as of 2015, was 4.7 million inhabitants [55]. Within its orography, the Sierra Madre Oriental, Sierra San Miguel, the hills of Topo Chico, La Silla, and Las Mitras are prominent.

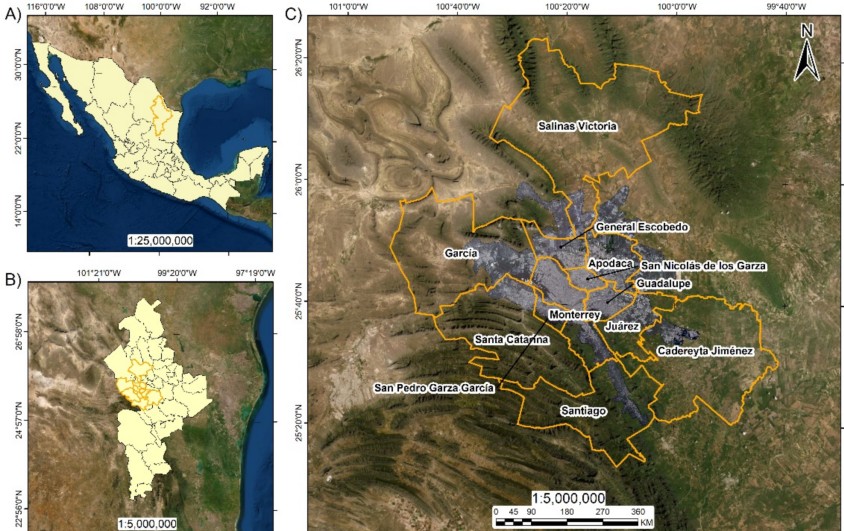

**Figure 1.** Study area location. (**A**) State of Nuevo Leon within the Mexican Republic; (**B**) Monterrey Metropolitan Area (MMA) within the state of Nuevo Leon; (**C**) orthomosaic of WorldView-2 coverage of the MMA.

The methodological workflow is shown in Figure 2, where the methodology is divided into three sections: data preprocessing, CNN model implementation, and evaluation of semantic segmentation of UGSs. For this study, we used a UGS definition based on the "Regulation of Environmental Protection and Urban Aesthetics of Monterrey" [56]. This document describes UGSs as land surfaces containing vegetation, gardens, groves, and complementary minor buildings for public use within the urban area or its periphery. Input label polygons for the CNN models were obtained from three sources, the UGS database of the National Geostatistical Framework (2010) of the National Institute of Statistics and Geography (INEGI), the collaborative Open Street Maps (OSM) project [57], and the database of median strips of the MMA arranged by the Department of Geomatics of the Institute of Civil Engineering of the UANL [58].

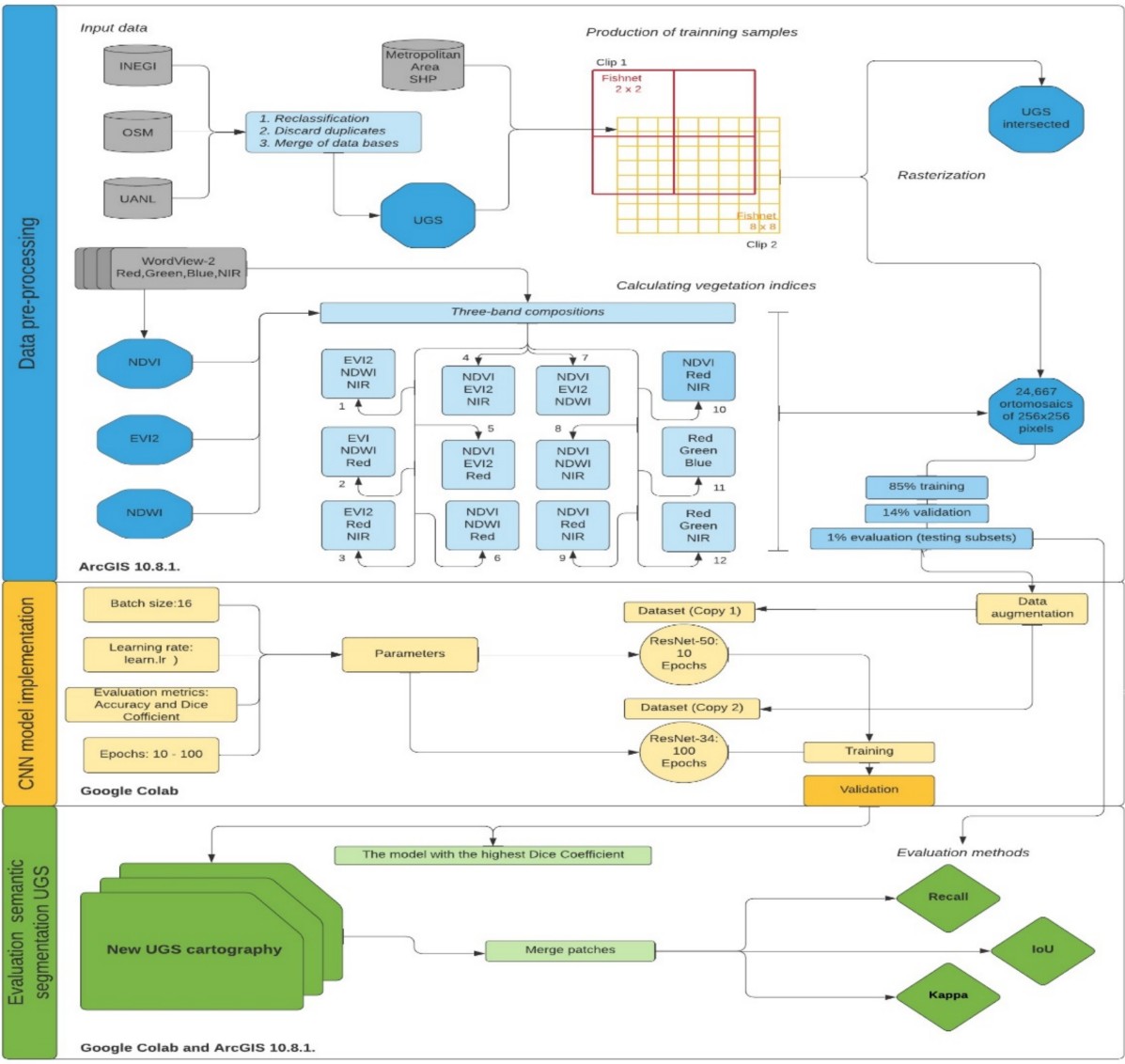

**Figure 2.** Summary of the current methodological process for semantic segmentation of UGSs using deep learning. Data preprocessing in blue, CNN model implementation in yellow, and evaluation of semantic segmentation of UGSs in green. Input data in gray, and all the processes appear in light blue, yellow, and green. Abbreviation: Urban Green Spaces (UGS), Convolutional Neural Networks (CNN).

### 2.2. Input Data

An orthomosaic of the MMA with a 0.5 m pixel resolution was used for the classification of UGSs (Figure 1). It was generated from nine WorldView-2 (WV2) satellite images

obtained between June and October 2017. The spectral information it contained includes the red, green, blue (RGB) and near-infrared (NIR) bands in the ranges of 630–690 nm, 510–580 nm, 450–510 nm, and 770–895 nm, respectively.

### 2.3. Data Preprocessing

Information of the three original databases was reclassified based on the function and geometric structure of the UGSs to generate a common UGS database. The database includes polygons representing (1) median strips along streets and avenues, which are characterized by their elongated and narrow shapes; (2) residential gardens, which have pixels that correspond to vegetation managed by the municipality; (3) roundabouts, which have a round shape; (4) squares, which are spaces mostly used for recreation that maintain a symmetry and lack elements related to sports; (5) parks, which are embedded in residential areas, used for recreation and sports, and tend to be asymmetrical. Original classifications (some of them in Spanish) and their equivalent names after the reclassification process are shown in Table 1.

**Table 1.** Label reassignment on the reclassification process of the three databases.

| Database | Original Fields | | Reclassified Fields |
|---|---|---|---|
| Department of Geomatics (UANL) | NA [1] | | Median strips |
| INEGI | Geográfico [1] | Tipo 1 [1] | |
| | | Bordo [1] | Median strips |
| | Camellón [1] | Camellón [1] | Median strips |
| | | Glorieta [1] | Roundabouts |
| | | Área verde [1] | Parks |
| | Plaza [1] | NA [1] | Squares |
| | Instalación deportiva o recreative [1] | Parque [1] | Parks |
| | | Jardín [1] | Residential gardens |
| OSM | Leisure Playground Park Common | | Parks Parks Parks |

[1] Original data in Spanish.

Polygons that presented overlap were discarded with the employment of ArcMap "select by location" tool. The three reclassified databases were merged to produce the input shapefile for a rasterization process. The resulting product had a resolution of 0.5 m and was carried out for the generation of the final label raster. The pixel values determined the presence or absence of UGSs corresponding to median strips, roundabouts, parks, squares, and residential gardens. An additional sixth class named non-UGS was added to cover background pixels.

With the use of the green, red, and NIR bands, normalized difference vegetation index (NDVI) [59,60], two-band enhanced vegetation index (EVI2) [61,62], and normalized difference water index (NDWI) [63] were calculated using Equations (1)–(3), respectively.

$$\text{NDVI} = \frac{(NIR - Red)}{(NIR + Red)} \tag{1}$$

where *NIR* represents the near-infrared channel and *Red* is the red channel.

$$\text{EVI2} = 2.5 \frac{(NIR - Red)}{(NIR + 2.4 * Red + 1.0)} \tag{2}$$

where *NIR* represents the near-infrared channel and *Red* is the red channel.

$$\text{NDWI} = \frac{(Green - NIR)}{(Green + NIR)} \tag{3}$$

where *Green* represents the green channel and *NIR* is the near-infrared channel.

This study used 12 three-band compositions to determine their potential for the segmentation of UGSs. The bands used for the combinations were the produced indices (NDVI, NDWI, EVI2) and the spectral single bands NIR, red, green, and blue obtained from the original WV2 data (Figure 1).

As part of the process, 24,667 orthomosaics with dimensions of 256 × 256 pixels were produced from the label raster and each of the 12 three-band compositions was generated for the MMA. To obtain those orthomosaics, the original compositions and their respective label rasters were clipped using first a 2 × 2 mosaic fishnet (Clip1) with resulting orthomosaics that cover an area of 1336.64 km². Using the results of Clip1, a second fishnet of 8 × 8 mosaics (Clip2) was applied to each section to obtain 135 segments of 167.08 km², with 50 produced for the quadrant of the cardinal NE position, 43 for the NW, 9 for the SE, and 33 for the SW. Both fishnets developed for the generation of training samples from the MMA orthomosaic and UGS labels are shown in Figure 3A. Subsequent split raster process was performed in ArcMap, ArcGIS v10.8.1 software, for each of the quadrants previously generated, and over 24,000 orthomosaics were obtained for each of the three-band combinations as well as their equivalent ground truths. All the data had a spatial resolution of 0.5 m. The data were divided in a proportion of 85% for training and 14% for validation [64]. An additional 1% of the information was used for the evaluation of the model. The results were hosted on Google Drive's cloud storage service for later use through the Google Colab platform, which provided a Tesla P100 PCIe 16 GB GPU.

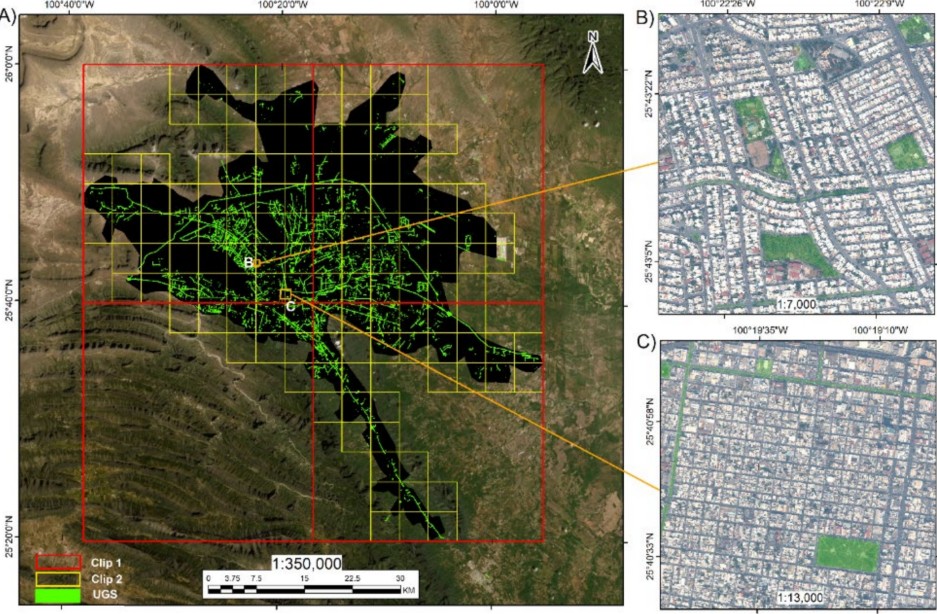

**Figure 3.** Production of training samples. (**A**) A 1:350,000 scale map with the result of data homogenization and rasterization of UGSs (light green polygons) with two raster extraction processes shown in red mosaics fishnet (Clip 1) and yellow fishnet (Clip 2). (**B**) Irregular geometry 1:7000 scale map of UGSs (light green polygons) extracted from Clip 2 and (**C**) Irregular geometry 1:13,000 scale map of UGSs (light green polygons) extracted from Clip 2.

### 2.4. CNN Model Implementation

Twenty-four semantic segmentation models were implemented via CNN, two for each band composition generated. ResNet-34 and ResNet50 encoders pre-trained by the

ImageNet dataset [65] were used in a dynamic U-Net architecture implemented in the fastai deep learning library [66]. This library works with the Python language and the PyTorch library [67] as a backend. Figure 4A shows the architecture for the model based on the pre-trained ResNet-34 encoder. Each model received 256 × 256 pixel images as input. ResNet network architectures integrate connection jumps (Figure 4B) that avoid the leak gradient problem present in other types of networks [68]. This helps to maintain performance and precision despite increases in the number of training layers [69]. At the point of the greatest compression in the FCN, a decoder was attached that follows the principle of the U-Net architecture to finally obtain an output equal in size to the input images.

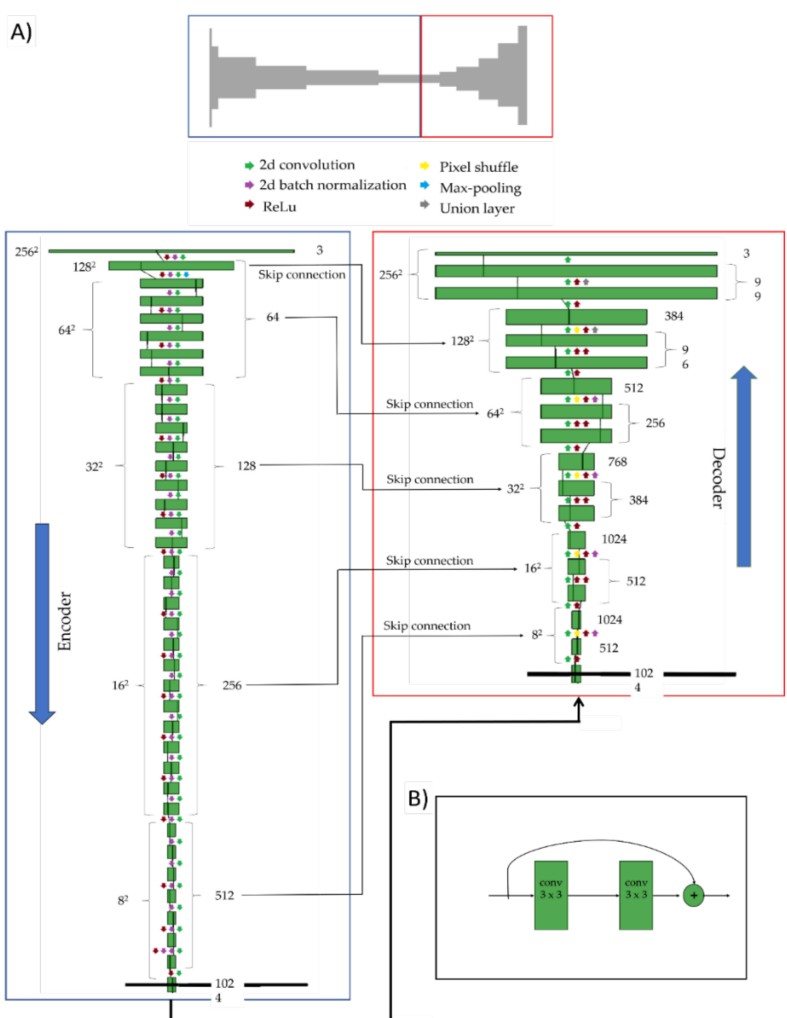

**Figure 4.** Dynamic U-Net model used for semantic segmentation. (**A**) The encoder consists of a ResNet-34 into which the orthomosaic and UGS labeling information is integrated. The numbers on the left of the graphs are the sizes of the input and output images, and the sizes of the activation maps. The numbers on the right are the channels/filters. (**B**) Building blocks used in the encoder section of the model.

Preliminary tests consisted of trial and error based on the limited literature related to UGS segmentation using deep learning models [38,70]. According to the capabilities of the system, a batch size of 16 samples was assigned. Data augmentation included (1) transformations with image turns at different angles, finding a 50% probability of being horizontal or vertical; (2) random symmetrical deformations with values of 0.1 magnitude; (3) random rotations with angles of 20°; (4) changes in the focus of images, up to 200%;

and (5) changes in light and contrast by a factor of 0.3. These techniques generated transformations for each epoch within the models and increased the size of the training samples by 60 times. Examples of image transformations can be observed in Figure 5.

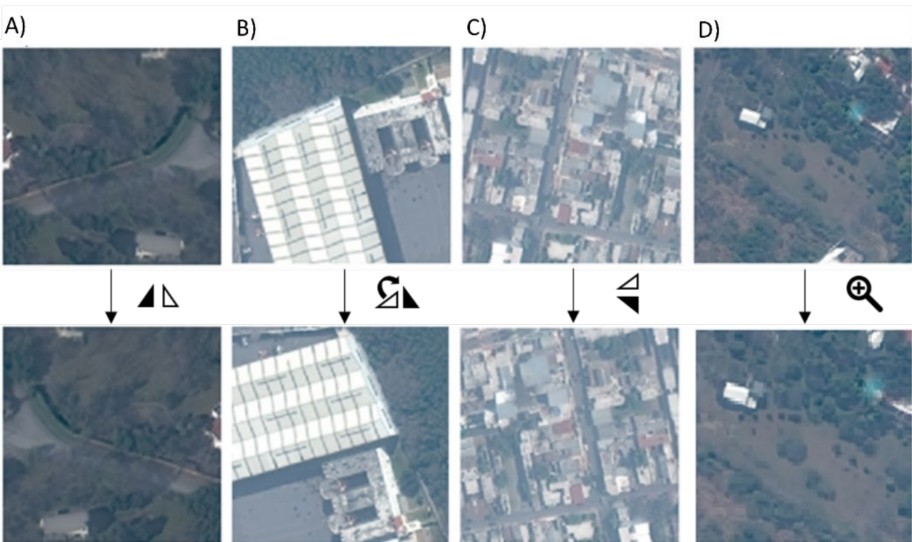

**Figure 5.** Example of images produced by data augmentation. (**A**) Horizontal flip; (**B**) random rotations; (**C**) vertical flip; (**D**) change in focus.

The accuracy score is an evaluation metric that quantifies the percentage of correctly classified pixels made by the predictions of the model [71]. It is calculated by Equation (4).

$$\text{Accuracy Score} = \frac{TP + TN}{TP + TN + FP + FN} \tag{4}$$

where *TP* represents the true positives, *TN* is the true negatives, FP is the false positives, and FN is the false negatives.

In semantic segmentation, the loss function metric is an algorithm used to evaluate the difference between training results and labeled data. To determine the most appropriate loss function metric for our data, we considered their spatial characteristics. As UGS represents a small portion of pixels at the metropolitan level, most cities are covered by built-up areas occupied by streets, buildings, and other impervious surfaces. This configuration causes an imbalance of classes that could produce errors and bias towards the background class that covers most of the area of interest. Semantic segmentation studies that used deep learning [72–74] have proved that the Dice coefficient or F1 score [75] is a loss function adequate for these kinds of problems. The Dice coefficient is calculated according to Equation (5).

$$\text{Dice Coefficient} = \frac{2|I_{GT} \cap O_{SEG}|}{|I_{GT}| + |O_{SEG}|} \tag{5}$$

where $I_{GT}$ is the input ground truth and $O_{SEG}$ is the output segmentation.

A Dice coefficient of 0 indicates that there is no overlap between the data, whereas a value of 1 means that the data has total overlap [76]. Because the input data consisted of three-band composed images, the Dice coefficient was computed for each class and then averaged via arithmetic mean through the fastai implementation [77]. The optimal learning rate for each model was defined using the *learn.lr_find()* method present in the fastai library [66]. This hyperparameter increases the learning rate from an exceptionally low value to the point where the loss gradient decreases [78]. Each ResNet34 model had 100 epochs to test the functionality of the implementations, this value was established according to the literature [79,80]. ResNet50 was implemented using 10 epochs due to

reported Google Colab limitations on GPU, RAM, and session time availability. Mean, standard deviation, and confidence intervals of 95% were produced for the models to determine the statistically significant differences between the Dice coefficient calculated for each three-band combination.

### 2.5. Evaluation of Semantic Segmentation of UGSs

The model with the highest Dice coefficient was evaluated using the additional 1% testing subset taken from the original information. This subset, which was produced using the semantic segmentation, results in footprints used to obtain vector information corresponding to UGS polygons. Vectorization was implemented using Solaris library (https://solaris.readthedocs.io/, 26 November 2020) on Google Colab. The generated polygons were downloaded and then analyzed in ArcGIS to evaluate the model. The evaluation data contained 68 mosaic samples (Figure 6) with a total coverage of 32.08 hectares (ha). These samples permitted the evaluation of the effectiveness of the CNN with images different from those of the training and validation sets.

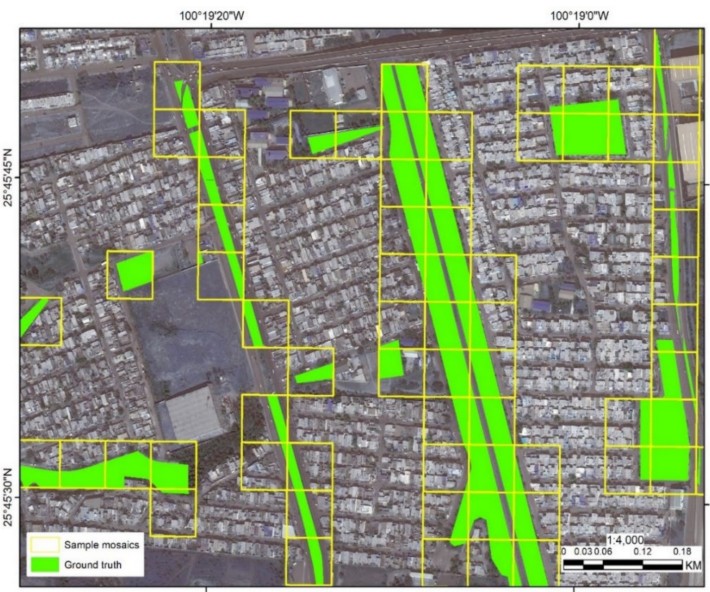

**Figure 6.** The ground truth coverage and the testing subset with 68 mosaics used to evaluate the CNN model.

As part of the evaluation, the intersection over union (IoU) was calculated. This metric computes the amount of overlap between the predicted polygons and the ground truth data [81] (Equation (6)).

$$\text{Intersection over Union } = \frac{|I_{GT} \cap O_{SEG}|}{|I_{GT} \cup O_{SEG}|} \tag{6}$$

where $I_{GT}$ is the input ground truth and $O_{SEG}$ is the output segmentation.

The recall analysis was obtained for the evaluation. This metric calculates the proportion of positives identified correctly as shown in Equation (7).

$$\text{Recall } = \frac{TP}{TP + FN} \tag{7}$$

where $TP$ represents the true positives and $FN$ is the false negatives.

The overall user and producer accuracies and the kappa coefficient were processed for the accuracy assessment of the evaluation data. This index of agreement is obtained through the computation of a confusion matrix with errors of omission and commission between classified maps and ground truth data; a kappa coefficient of 1 represents a perfect

agreement, and a value of 0 indicates that the agreement is not as it was expected by chance [82,83]. For the calculation of the index, 2000 random points were generated by stratified random sampling and were labeled and verified against the reference data. The kappa coefficient (Equation (8)) is computed as follows:

$$\text{Kappa Coefficient} = \frac{N \sum_{i=1}^{r} x_{ii} - \sum_{i=1}^{r}(x_{i+} * x_{+i})}{N^2 - \sum_{i=1}^{r}(x_{i+} * x_{+i})} \tag{8}$$

where $r$ represents the number of rows and columns in the error matrix, $N$ is the total number of pixels, $x_{ii}$ is the observation in row $i$ and column $i$, $x_{i+}$ is the marginal total of row $i$, and $x_{+i}$ is the marginal total of column $i$.

## 3. Results

### 3.1. Data Preprocessing

The final dataset used as model input is shown in Table 2. It presents numbers of polygons, the area covered by the UGS classes within the MMA, and the proportions of area. The UGS pixel ratio was 3.003%. The data augmentation technique helped to improve the amount of supporting information for the training process by increasing the number of mosaics from 24,667 to 1,480,020 mosaics. With this increase, the CNN improved the learning process.

**Table 2.** Results of the homogenization of UGS databases.

| UGS | Polygons | UGS Area (m$^2$) | Proportion (%) |
|---|---|---|---|
| Median strips | 19,869 | 1,141,179 | 0.843 |
| Residential gardens | 1818 | 463,314.5 | 0.342 |
| Roundabouts | 61 | 810 | 0.001 |
| Squares | 58 | 14,076 | 0.010 |
| Parks | 2861 | 2,446,925 | 1.807 |
| TOTAL | 24,667 | 4,066,304.5 | 3.003 |

Parks represented the most prominent type of UGS with 1.8% coverage of the MMA (Table 2). Median strips represented 0.84% cover. Residential gardens represented 0.34% cover. The classes with the lowest coverage were squares and roundabouts with 0.01% and 0.001%, respectively.

### 3.2. Semantic Segmentation of UGSs

The highest Dice coefficient and accuracy results of the semantic segmentation for each of the 12 three-band compositions are presented in Table 3. NDVI–red–NIR composition achieved the best results using ResNet34 encoder with a Dice coefficient value of 0.5748 and an accuracy of 0.9503. Red–green–blue composition achieved the best results using ResNet50 with a Dice coefficient of 0.4378 and an accuracy of 0.9839. In contrast, EVI2–NDWI–NIR composition had the lowest values for both encoders. For the ResNet34 encoder model, the mean Dice coefficient was 0.49, the standard deviation was 0.09, and the statistical significance using 95% confidence intervals ranged from 0.42 to 0.55. For the ResNet50 encoder model, the mean Dice coefficient was 0.42, the standard deviation was 0.58, and the statistical significance using 95% confidence intervals ranged from 0.28 to 0.36.

Figure 7 illustrates the behavior of the training and validation process for the highest Dice coefficient for both encoders. As observed, the ResNet34 learning process extended to the 100 epochs (fluctuating between 0.45 and 0.63) and presented its peak at the 83rd epoch, reaching a Dice coefficient of 0.5748. In contrast, the ResNet50 learning process activity occurred during the first 10 epochs, reaching the best Dice coefficient of 0.4378 on the 4th epoch.

**Table 3.** Semantic segmentation model validation results for UGSs in VHR satellite images.

| Band Compositions | ResNet34 | | ResNet50 | |
|---|---|---|---|---|
| | Dice Coefficient | Accuracy | Dice Coefficient | Accuracy |
| EVI2–NDWI–NIR | 0.1940 | 0.8853 | 0.2231 | 0.9065 |
| EVI2–NDWI–Red | 0.4961 | 0.9337 | 0.2543 | 0.9147 |
| EVI2–Red–NIR | 0.5113 | 0.942 | 0.3199 | 0.9145 |
| NDVI–EVI2–NIR | 0.5307 | 0.9437 | 0.2698 | 0.9074 |
| NDVI–EVI2–Red | 0.5021 | 0.9452 | 0.3356 | 0.9227 |
| NDVI–NDWI–Red | 0.5248 | 0.9433 | 0.3187 | 0.9115 |
| NDVI–EV2–NDWI | 0.4617 | 0.9347 | 0.3548 | 0.9249 |
| NDVI–NDWI–NIR | 0.4886 | 0.9377 | 0.2763 | 0.9369 |
| NDVI–Red–NIR | 0.5748 | 0.9503 | 0.3149 | 0.9004 |
| NDWI–Red–NIR | 0.5702 | 0.9505 | 0.3610 | 0.9200 |
| Red–Green–Blue | 0.4638 | 0.9792 | 0.4378 | 0.9839 |
| Green–Red–NIR | 0.5193 | 0.9547 | 0.3663 | 0.9322 |

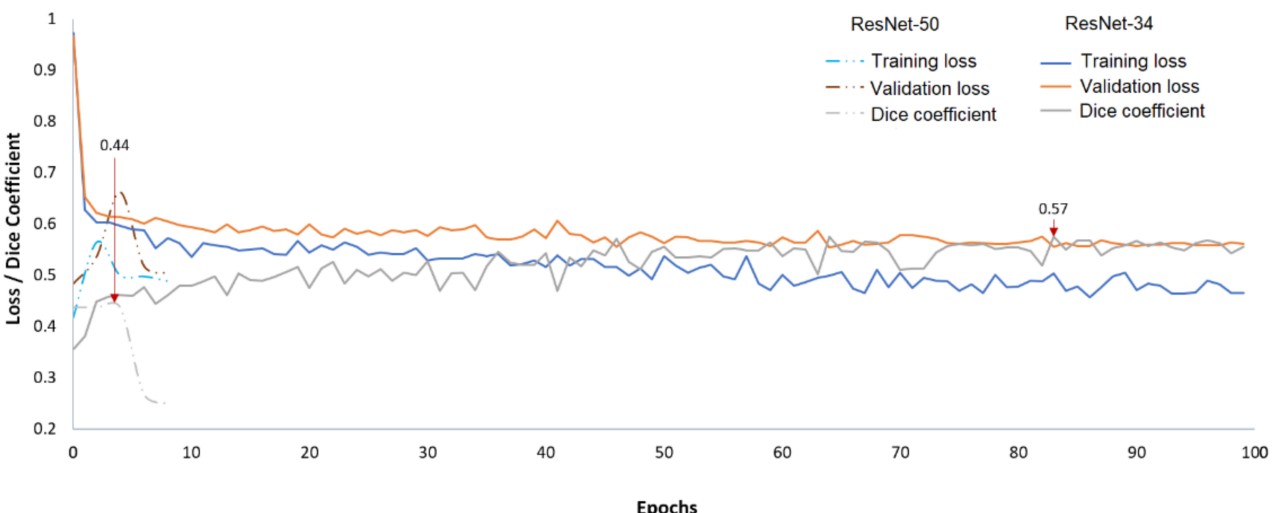

**Figure 7.** Plot of training loss, validation loss, and Dice coefficient for both encoders.

The best segmentation process is represented by the lowest loss value. An example of this is shown in Figure 8A–D, where NDVI–red–NIR composition using ResNet34 reflects how the learning process increases from a loss of 1.14 to 0.77. This behavior is also observed for the RGB combination using ResNet50 where the loss was from 1.47 to 0.84 (Figure 8E–H).

IoU metric was 0.75 for the evaluation of the NDVI–red–NIR composition. This was calculated using the polygons presented in Figure 9A. The recall analysis revealed that the ground truth data had an overlap of 96.07% with the predicted data and the proportion of the overlapping polygons corresponding to the predicted data was 80.04% (Figure 9B).

Results of the confusion matrix and kappa coefficient produced for the evaluation dataset are shown in Table 4. Both the ground truth and the predicted data contained polygons corresponding to parks and median strips classes. The kappa coefficient was 0.94, and the overall accuracy calculated was 0.97. The user accuracy was 1 for the parks and 0.96 for median strips, and the producer accuracy was 0.92 for the parks and 1 for median strips.

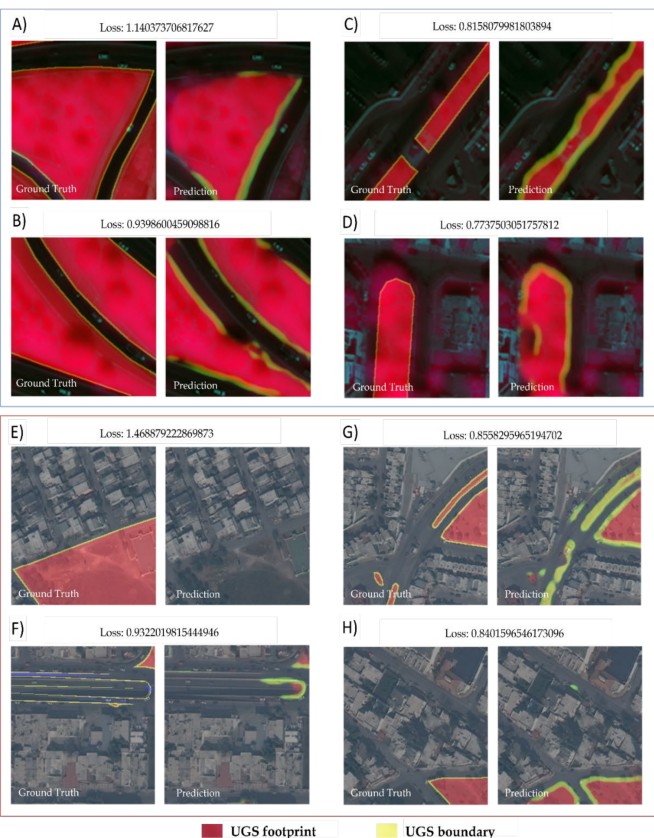

**Figure 8.** Progression of the segmentation process of UGSs for both encoders. (**A–D**) Samples obtained using an NDVI–red–NIR composition from ResNet34 encoder. (**E–H**) Samples obtained using a red–green–blue composition from ResNet50 encoder. The information is presented at the same scale; each square surface is 1.63 hectares (ha).

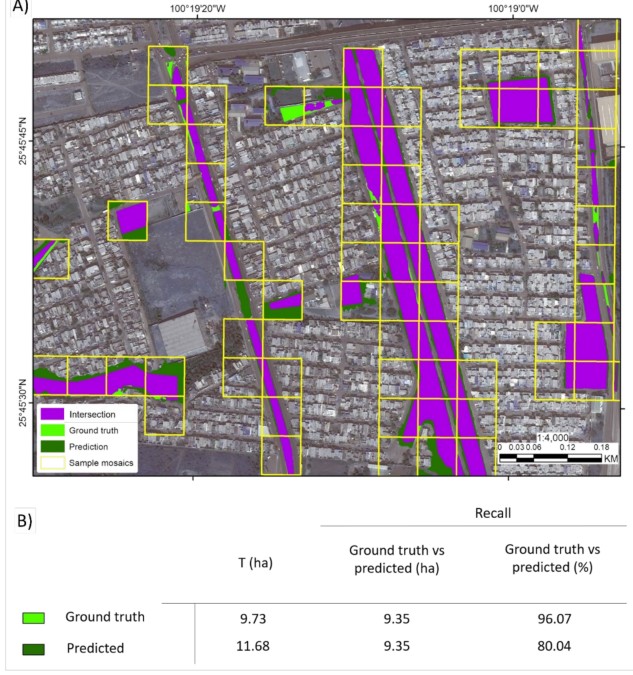

**Figure 9.** Model assessment. (**A**) The overlap coverage between the polygons produced from the CNN model and the ground truth data. (**B**) Recall analysis. T is the total coverage. Ground truth vs predicted columns show the area in ha and proportion of positives identified correctly.

**Table 4.** Confusion matrix and kappa coefficient.

|  | Parks | Median Strips | Total | User Accuracy | Kappa Coefficient |
|---|---|---|---|---|---|
| Parks | 684 | 0 | 684 | 1 | 0 |
| Median strips | 58 | 1258 | 1316 | 0.96 | 0 |
| Total | 742 | 1258 | 2000 | 0 | 0 |
| Producer accuracy | 0.92 | 1 | 0 | 0.97 | 0 |
| Kappa coefficient | 0 | 0 | 0 | 0 | 0.94 |

## 4. Discussion

The two automated methods tested for the identification of individual UGS polygons at the metropolitan level generated new databases that provide useful information, including geometry, condition, and spatial attributes, for the decision-making process regarding these important open public spaces. Updated databases improve national inventories with detailed geospatial and geometric information of UGSs, which is important for assessing their distribution and management. Updated and accurate UGS information, however, is difficult to acquire or access, especially in developing countries with no access to VHR imagery. Latin American countries lack this kind of information, where the UGS inventories are typically based on photo-interpretation techniques and depend on the user experience.

The information produced with this methodology can be used in conjunction with demographic information to analyze the accessibility and connectivity of UGSs at the metropolitan level. While this is a portion of the information needed for the quantification of the achievement of the SDG 11.7, the generation of a similar database considering non-green public spaces should also be contemplated to cover the analysis of both elements.

The methodology of segmentation of UGS polygons at the city level proposed in this work will allow effectively updating this information for urban spaces in Mexico. The method includes the typical UGS classes, such as median strips, roundabouts, parks, squares, and residential gardens present in every metropolitan area. OSM and INEGI open access databases used in this research are available for all of Mexico. This information was complemented by information produced by the local university through a metropolitan project funded by the state government, proving the importance of integrating multilevel governance (or institutions) to enhance and update geospatial data such as UGS inventories.

Methods to increase data representation are a necessity when VHR imagery is limited. In this work, data augmentation techniques using simple strategies showed their effectiveness by providing over 1 million additional orthomosaics. The variations that occurred in each of these image transformations helped to reach a more complete training set representing the complexity occurring in the study area due to temporal or environmental disparities.

The prospected combinations produced by four-band VHR imagery and their implementation using two encoders allowed the assessment of 24 segmentation models. This kind of modeling is only possible with a high computation capability. When that is not available, other options such as deep learning processing cloud services (e.g., Google Colab, Amazon AWS, Microsoft Azure) can be implemented.

According to the semantic segmentation model validation results and its statistical analysis, there is a significant difference (95%) between the dice coefficient of the different band combinations in both models. The best three-band combination for the semantic segmentation of UGSs is NDVI–red–NIR when using ResNet34 encoder and red–green–blue when using ResNet50 encoder. A future analysis regarding the learning process could help to identify the learning patterns and the influence of each band within the models by using interpretability, representation learning, and visualization methods [84].

A large difference between the validation accuracy and the dice coefficient was observed in Table 3. This variance is associated with the data imbalance caused by background non-UGS pixels. As the non-UGS class has the highest number of pixels in the analyzed orthomosaics, the accuracy of the model is high as it is quantifying a high percentage of correctly classified pixels for the entire area. The dice coefficient is a more reliable

parameter in semantic segmentation processes with data imbalances because it reflects a metric based only on the segmented classes.

Semantic segmentation studies using similar approaches to map urban tree coverage, buildings, and roads [85–87] reported dice coefficients of 0.94, 0.84, and 0.87, respectively. The result in this study is lower (0.57), which may be related to the complexity of mapping UGS polygons. The referenced studies focus on the segmentation of classes that represent the city coverage; however, this research seeks to segment the pixel class and also several types of geometry. Additionally, UGS polygons are composed of a mix of pixels representing not only vegetation but also other kinds of infrastructure, such as sidewalks and playgrounds, which decrease the certainty for the segmentation process.

The kappa coefficient produced in the accuracy assessment of the evaluation data indicated a strong agreement between the predicted polygons and the ground truth data. With a value of 0.94, the kappa coefficient was similar to high accuracy results obtained in recent studies related to UGS mapping methods [88,89]. This indicates that the methodology used in this study is accurate for extracting and updating geometrical UGS databases at the metropolitan level.

## 5. Conclusions

This study evaluated two deep learning model techniques for semantic segmentation of UGS polygons with the use of different CNN encoders on the U-Net architecture to improve the methodology of UGS cartography. The models have the capability to detect patterns for all types of UGSs reported in Mexico, even with a high variation in shape or size, and to segment hundreds of thousands of polygons that represented 3% of the total MMA.

Results demonstrate that this methodology is an accurate digital tool for extracting and updating geometrical UGS databases at the metropolitan level (Dice coefficient of 0.57, recall of 0.8, IoU of 0.75, and kappa coefficient of 0.94). The implementation of these models could update UGS inventories necessary to assess urban management as cities grow or change. This methodology produces UGS geospatial data that are essential for quantifying the accomplishment of the SDG 11.7 regarding green spaces. This information in combination with demographic data could be used to elaborate UGS accessibility maps necessary to assess UGS accessibility. This new cartography may improve urban management for the conservation of natural resources and the environmental services they provide, as well as making their maps more accessible to urban residents and decision-makers.

**Author Contributions:** Conceptualization, F.D.Y.; data curation, R.E.H.; formal analysis, R.E.H. and F.D.Y.; funding acquisition, F.D.Y.; investigation, R.E.H. and F.D.Y.; methodology, R.H.G. and F.D.Y.; project administration, F.D.Y.; software, R.E.H.; supervision, F.D.Y., R.A.C.G., D.F.L.-G., V.H.G.C., A.L.F.F., and H.d.L.G.; validation, F.D.Y. and D.F.L.-G.; visualization, R.H.G. and F.D.Y.; writing—original draft, R.H.G. and F.D.Y.; writing—review and editing, D.F.L.-G., V.H.G.C., A.L.F.F., H.d.L.G., R.A.C.G. and A.V.M. All authors have read and agreed to the published version of the manuscript.

**Funding:** This research was funded by the National Council of Science and Technology, grant number 559018 and grant for international research (Foreign Mobility 2019-1).

**Acknowledgments:** The authors extend their sincere thanks to Gustau Camps-Valls, researcher of the Image Processing Lab of the University of Valencia, who guided Roberto Huerta in neural network subjects through a stay in the Image Processing Laboratory. Map data are copyrighted by OpenStreetMap contributors and available from https://www.openstreetmap.org, (accessed on 26 November 2020). The authors thank William D. Eldridge for his support and comments.

**Conflicts of Interest:** The authors declare no conflict of interest.

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
