# Peer review of "Mapping Urban Green Spaces at the Metropolitan Level Using Very High Resolution Satellite Imagery and Deep Learning Techniques for Semantic Segmentation"

_remotesensing, doi:10.3390/rs13112031_

Round 1
Reviewer 1 Report
Dear Authors,
Your work on “Mapping urban green spaces at the metropolitan level using very high-resolution satellite imagery and deep learning techniques for semantic segmentation” shows how deep-learning techniques can be used to map urban planning and environmental relevant information for a large metropolitan area. The paper is interesting, in particular, the scale of the analysis and the methods used to generate training data from an available data repository. However, the paper would require an improvement in several aspects:
- You refer in your work to the “SDG 11.7.1.b” -> I do not understand what the b stands for the SDG indicator on urban green spaces is: “11.7.1: Average share of the built-up area of cities that is open space for public use for all, by sex, age and persons with disabilities”. When referring to this indicator, you would also need to discuss that with the method, you only measure green spaces, not open space, and you do not produce information related to whether people (demographic information required) can access them.
- The experimental set-up raises one important question – why are you not comparing the different band combinations also to a combination that uses the original bands, which include the NIR band. It seems odd when mapping green spaces to use a set with only VIS bands. Furthermore, you could modify the network to use 4 bands as input.
- The text would require a bit of revision to improve the language and further clarify results.
Some detailed comments:
Introduction:
2nd paragraph: Please keep in mind that not only the availability of USG is important, but also the proximity to UGS!
3rd paragraph: “UGS a major challenge for government administrations in metropolitan areas with extensive industrial development” -> not only because of industrial developments – but also other developments put pressures on UGS!
Last paragraph: “The Sustainable Development Goal (SDG) 11.7.1.b indicate the necessity to map and calculate the total area of open public space available” -> please correct the “b” and please keep in mind that open spaces are not only green spaces and that access is an important aspect that is difficult to assess from EO-data!
Methods:
Figure 2: The link between the training and output of deep learning is not clear! They are not connected in your diagram!
Section 2.3: The statement “A double ground truth clipping process with 2x2 and 8x8 fishnets was employed” is not clear – could you please improve/clarify!
Section 2.3 “The data was divided in a proportion of 85% for training and 15% for validation [61]. An additional 1% of the information was used as a testing subset” -> This seems odd a total of 101%?
Figure 3: I would suggest using outlines or transparency to show the green areas in the maps B and C!
Result:
Section 3.1: “Due to the elevated levels of urbanization, the metropolis concentrates numerous streets and avenues that are accompanied by median strips” -> the sentence reads a bit odd – do you mean rapid urbanization? And I do not understand the relationship between urbanization and median strips?
Table 2: I would suggest to have here areas, not pixels!?
Table 3: is this the result of validation or testing? I guess validation – I would add this to the caption text.
Table 3: what causes the large difference between the Dice Coefficient and Accuracy for the VIS bands? Please explain/discuss!
Figure 9 and related text – what are the uncertainties of the ground truth? B) I do not understand the presented statistics? The associated text does not clarify it!
Discussion:
Please revise this statement – it reads odd “When VHR imagery is limited ingenious proposals….”
Conclusions:
Please be more specific about the scope (including limitations) of contributing to the SDG 11.7.1 indicator!
Author Response
Dear reviewer 1,
We appreciate the time you invested in this revision. All your comments were very accurate and we tried to follow them and solve them in our manuscript or figures, please see our responses to each of them in the attached file.
Best,
The authors

Reviewer 2 Report
Dear Authors,
Thank you for choosing Remote Sensing for your interesting work. The proposed approach is apparently gaining some meaningful insights. However, the study has gained some interesting insights but most of the EO community compares the accuracy measures using standard assessment estimates. I have seen the way how you assess deep learning algorithm performances but I can hardly compare it with a typical land-cover classification accuracy. Can you translate or try to use Kappa or other similar widely used metrics in addition to the ones you have reported? Thank you in advance for your kind cooperation!
Kind regards,
Reviewer
Author Response
Dear reviewer 2,
We appreciate the time you invested in this revision. Please see our responses to your comments in the attached file.
Best,
The authors

This manuscript is a resubmission of an earlier submission. The following is a list of the peer review reports and author responses from that submission.
Round 1
Reviewer 1 Report
In this manuscript, the authors present a study involving the development of the U-Net based semantic segmentation model for identifying urban green spaces (UGS). The paper is well written and the study seems like a valid contribution in the application domain of deep semantic segmentation. However, from a technical standpoint, the manuscript lacks novelty and even the application of the existing technologies could be significantly improved. For those reasons, I must advise rejecting the current manuscript.
These are my more detailed remarks:
Line 24: A common way to present results of a segmentation model is to use Intersection over Union (IoU) metrics and train it using the dice loss function.
Line 135: A typical method for evaluation is double cross-validation, i.e. splitting the dataset into training, validation, and testing subsets. The validation dataset is then used to pick the best performing model trained on the train set, while the testing dataset is used solely to evaluate the model and present results.
Line 144: The term "dynamic U-Net" is not introduced. The way you introduced it sounds like it is some kind of your advancement to U-Net architecture, but I cannot find evidence for that in the manuscript (using different CNN encoders in the U-Net is very common). I do not see whether you used a pretrained ResNet-34 encoder? Also, trying at least ResNet-50 encoder and comparing results would be nice.
Line 162: The image does not show skip connections that are a key characteristic of U-Net architecture.
Line 173: Having a figure illustrating different data augmentation transformation would be nice.
Line 176: TP + T shouldbe TP + TN
Line 184: Could you be more specific about the loss function used for the training?
Line 250: As I already stated, IoU should be used as the metrics to present results. It should be also stated are the results mean of partial IoU metrics for each image patch or they are calculated on the global scale? In my opinion, the results obtained on 256x256 patches are informative only to a certain degree. You may use the trained model to predict patches and then fuse those results to predict output for a certain area.
Line 323: What do you mean by "satisfactory performance of the semantic segmentation U-Net model"? You need to justify this claim.
Author Response
Dear reviewer,
We appreciate the time you kindly invested in this paper.
Please download and read the attached file with our responses.
Best regards,
Fabiola Yépez

Reviewer 2 Report
Dear authors,
Your manuscript on “Mapping urban green spaces integrating very high resolution satellite imagery and a semantic segmentation dynamic U-Net model” shows the potential of a very popular FCN to map urban green spaces and links it to the municipal information needs (often data on green spaces are not available or complete). At present, the paper has several major shortcomings and would require careful improvement. Some areas of improvements are the following:
- Introduction:
- When starting the main text, I would suggest introducing the acronym again: Urban green spaces
- As part of the introduction, I am missing a comprehensive overview of previous publications done on mapping green urban spaces with machine learning: e.g., https://doi.org/10.3390/ijgi8100463 or https://doi.org/10.3390/rs12071144 or https://doi.org/10.3390/rs12223845 or https://doi.org/10.1080/19479832.2020.1749142
- Therefore, also the research gap and your scientific contribution need to be better clarified.
- I would suggest linking the mapping of urban green spaces also to the data needs as part of the SDG 11.7.1 indicator.
- Methods and data:
- As part of the methodology, I am missing a clear definition of what are urban green spaces? As well as an overview of the different classes, their definitions, and why are these classes used?
- For the generation of training data, I believe you used OSM data, but it is not clear how the data was used to generate training data (e.g., did you improve the data)?
- Also, please reflect on the use of WV-2 data, they come with high data cost and might not be feasible for municipal use.
- In the section of line 127 etc – it seems that you only used the visible bands of WV-2? Why did you not use the NIR bands which are very important for vegetation cover mapping? (+ explain the rational for converting to 8 bit).
- “Line 167 Preliminary tests consisted of trial and error based on the limited literature related to UGS” – please add the references you refer to here!
- Equation 1 seems to have a mistake – what is T? I guess it should be TN?
- Equation 2 and results of F1 score – I do not understand why the results of your F1 scores are so much lower – how did you implement these measure. This would require some more specific explanation as part of the methodology section.
- I am missing information on how the polygons of UGS were generated from the the output of the FCN based result.
- Results:
- I do not understand this statement: “As seen, the UGS pixel ratio was 3.004%”?-> if you refer to the Total in Table 1 you need to match the numbers.
- The first sentence of section 3.2 is actually methodology.
- Table 2 and text in line 230 etc: why are you not including more epochs?
- Section of Line 238 etc: It is very difficult to follow which classes you take about – A,B,… I would suggest to include also the classes.
- Figure 5 and the related text – production of training data should be part of the methodology.
- Figure 7: please add a label to the y axis!
- Figure 8: no legend is included.
- Figure 9: the legend is not complete – several colours are not defined.
- Discussion:
- The discussion is difficult to follow and would require substantial improvements also relating it to recent literature and the application potential.
- Line 280 “the proportion of soil within the MMA” – this is the first time the class soil is mentioned? And I do not understand the entire sentence.
- Line 287 etc: You would need to explain the large difference between F1 and Accuracy. Also here it is not clear why this large difference exists and how the training data were prepared.
- Line 306 – I do not understand this statement “The best CNN model achievement was the automatic identification of polygons that were not present in the training information.” Need to be improved!
- Please add references (I think you overlook several recent works) “Although deep learning has already begun to be used for the analysis of UGS”. (For example https://doi.org/10.1016/j.envint.2019.02.013 or 1109/ICIVC47709.2019.8981007
- Conclusions:
- What do you mean by all types of UGS – for the missing introduction of the classes used it is not clear whether you capture all green urban spaces (e.g., urban agriculture)?
Author Response

(The authors gave the same response as above.)

Reviewer 3 Report
Comment: My major concern is about the contribution of the manuscript. The authors created a dataset and used a classical segmentation model. The authors should present and justify the novelty of their work at the Introduction.
Comment: Why indices such as NDVI, NDRE, etc were not used?
Comment: About the structure of the manuscript
- The authors should include the reminder of their work at the end of the Introduction
- A new section should be included after the Introduction presenting similar works and how this work is differenced.
Comment: Have the authors used any other backbone, except from ResNet-34??? It is interesting to see, how U-NET performs using several backbones (ResNet50, ResNet152, EfficientNet5, EfficientNet6, etc) and also examine how challenging is the segmentation problem.
Comment: The authors should also include IOU metric.
Comment: The conclusions section should be enlarged by at least one paragraph.
Comment: Line 94: “by” is in bold.
Comment: Line 176: Replace “T” with “TN”
Author Response

(The authors gave the same response as above.)

Round 2
Reviewer 1 Report
The authors significantly changed the manuscript to address the comments that I had, so my recommendation is to accept the manuscript for publication after providing the final version (without track changes) for one additional check.
Reviewer 3 Report
The presentation of the manuscript has not been improved in order to be worthy of publication.
- The contribution of the manuscript is limited.
- I do not agree with the authors in the utilization of RGB photos.
A combination which could easily provide them better performance could be NDVI-NDBI-NDWI which is commonly used in the literature. Additionally, there are even more advanced indices such as EVI (Enhance Vegetation Index).